# A Fast Point Cloud Recognition Algorithm Based on Keypoint Pair Feature

**DOI:** 10.3390/s22166289

**Published:** 2022-08-21

**Authors:** Zhexue Ge, Xiaolei Shen, Quanqin Gao, Haiyang Sun, Xiaoan Tang, Qingyu Cai

**Affiliations:** 1College of Intelligent Science, National University of Defense Technology, Changsha 410073, China; 2College of Information Engineering, Henan University of Science and Technology, Luoyang 471000, China; 3Department of Mechanical and Electrical Engineering, Changsha College, Changsha 410005, China; 4Hunan Sany Industrial Vocational and Technical College, Changsha 410129, China

**Keywords:** 3D object recognition, 3D pose estimation, point cloud, point pair feature, keypoint extraction, angle-adaptive judgment

## Abstract

At present, PPF-based point cloud recognition algorithms can perform better matching than competitors and be verified in the case of severe occlusion and stacking. However, including certain superfluous feature point pairs in the global model description would significantly lower the algorithm’s efficiency. As a result, this paper delves into the Point Pair Feature (PPF) algorithm and proposes a 6D pose estimation method based on Keypoint Pair Feature (K-PPF) voting. The K-PPF algorithm is based on the PPF algorithm and proposes an improved algorithm for the sampling point part. The sample points are retrieved using a combination of curvature-adaptive and grid ISS, and the angle-adaptive judgment is performed on the sampling points to extract the keypoints, therefore improving the point pair feature difference and matching accuracy. To verify the effectiveness of the method, we analyze the experimental results in scenes with different occlusion and complexity levels under the evaluation metrics of ADD-S, Recall, Precision, and Overlap rate. The results show that the algorithm in this paper reduces redundant point pairs and improves recognition efficiency and robustness compared with PPF. Compared with FPFH, CSHOT, SHOT and SI algorithms, this paper improves the recall rate by more than 12.5%.

## 1. Introduction

With the widespread usage of depth sensors, whether in pure 3D data applied to virtual reality or virtual reality fusion applied to augmented reality, it is necessary to identify and register 3D objects with different poses. As such, obtaining accurate and reliable 6D poses of objects through 3D data has received more and more attention. Over the last decade, several new forms of 3D object identification and pose estimation systems have been suggested. At present, it is mainly divided into the following categories: (1) The ICP algorithm proposed by Besl [1] registers the point cloud, but the ICP algorithm is prone to fall into the local optimal solution. As such, further scholars have studied this problem, [2,3,4] and other algorithms have improved algorithms based on ICP, which are more robust against false matches, avoid expensive nearest neighbor searches, and maintain the accuracy of the algorithm. (2) ICP is combined with other algorithms. Algorithms such as [5,6] combine ICP with other registration algorithms, which has advantages in registration accuracy, but take longer and aree not suitable for point cloud matching with less obvious features. The standard effect is not very good. (3) Feature-based matching methods, [7,8,9,10,11,12,13,14,15] aimed at feature extraction and description, and improve the accuracy of the registration algorithm through different feature extraction algorithms and descriptions, but the parameters of some algorithms should not be adjusted, and different point clouds need to adjust different parameters; the above feature methods are not robust when the point cloud is severely occluded. In order to solve this problem, Drost et al. [16] proposed a point pair feature algorithm (PPF), which combines the global and local advantages and can be used to quickly complete the point cloud feature description, pose voting, and solve the rough registration process of the rotation and translation matrices. (4) In addressing other aspects, Chang et al. [17] proposed a non-rigid registration algorithm by performing K-means clustering on two point clouds and constructing a connection relationship; Li Jun [18] proposed a point cloud registration algorithm based on extracting overlapping regions. In order to obtain high accuracy, these methods are highly descriptive, but the feature points extracted by the algorithms lack representativeness, are prone to wrong corresponding point pairs and are computationally complex.

Due to the wide application of the PPF algorithm, many PPF improvements have also emerged. Xiao et al. [19] proposed a plane constraint point pair feature (PC-PPF) algorithm, which introduces the convex hull algorithm to remove the plane points in the scene, reduces the number of descriptors, and improves the recognition speed. Based on PPF, D. Li et al. [20] proposed a multi-view rendering strategy to sample the visible model points, which is suitable for scenes with many planes. Hs-PPF was proposed by S. Hinterstoisser et al. [21], which introduced a novel PPF descriptor propagation strategy, which greatly improved the performance of Drost-PPF against sensor noise and background clutter. G. Wang et al. [22] proposed a novel voting strategy based on Hs-PPF to reduce the computational cost, but the recognition rate of the algorithm was lost. Yue et al. [23] proposed a fast and robust local point pair feature based on the thick and thin point cloud registration method (LPPF). Liu et al. [24] proposed PPF-MEAM, based on the descriptor B2B-TL, a method that uses multiple models to describe the target object. A method to improve the recognition rate and reduce the calculation time was proposed, but the applicability of this algorithm is relatively simple. Xu [25] proposed a recognition and localization method combining local image blocks and PPF, using deep convolution training images to improve the recognition effect of the algorithm, but it is too cumbersome to implement, and requires a large number of images for training to obtain better results. Bobkov [26] and others also combined convolutional network and PPF, and proposed a 4D descriptor convolutional neural network, which has strong advantages in high noise and occlusion scenes. Cui [27] et al. proposed Cur-PPF, which introduced the curvature information of point pairs to strengthen the feature description and improve the point cloud matching rate, but there are still useless model point pairs in the algorithm matching.

Different from previous algorithms, this paper proposes a 6D pose estimation method (K-PPF) based on keypoint pair feature voting, which is based on the point pair feature (PPF) [16] method that combines hash table and Hough voting. The keypoint sampling for the corner point extraction of the model object is used to improve the PPF sampling part. The point pair feature vector of the improved algorithm has differences and discreteness, which reduces redundant point pairs and can more completely express the 3D model’s characteristic information. Compared with the PPF algorithm, the recognition efficiency and robustness of the proposed algorithm are greatly improved, and it can achieve better results in less time. It can be used to quickly complete point cloud feature description, pose voting, and solve the coarse registration of rotation and translation matrices process, and finally combine point-to-plane ICP to derive the 6D pose. The main goals of this paper are as follows:On the basis of PPF, a keypoint extraction algorithm based on grid ISS sampling combined with curvature-adaptive sampling and angle-adaptive judgment is proposed. The algorithm has higher efficiency compared to the original PPF.Several sets of experiments are compared between K-PPF, PPF and other algorithms in public datasets, and the results prove the superiority and robustness of the K-PPF algorithm.

## 2. The Proposed Method

The K-PPF algorithm process is proposed based on the traditional PPF algorithm [16] and is used to quickly complete 3D object recognition and pose estimation. As shown in Figure 1, this algorithm is divided into offline stage and online stage.

### 2.1. Offline Stage

In the offline stage, a global model description is generated for the model point cloud and stored in a hash table. The PPF algorithm relies on the point pair feature description. Because PPF uses raster sampling, there are redundant and useless point pairs in this sampling method, and the sampling points cannot accurately describe some angular CAD models, so the algorithm’s robustness is weak, and some special shapes 3D objects may be computationally expensive and poorly recognized. So the sampling method should be improved.

Our goal is to make the point pair feature vector be invariant, discriminative, and discrete. The original point pair feature is invariant and repeatable to translation and rotation. In order to improve the distinguishability and discreteness of the point pair feature, a new key point extraction method is proposed. This keypoint extraction method can improve the discreteness of point pair feature and reduce the amount of calculation. Even in areas with large changes in edges, corners or surfaces, the key points can be well-extracted. For some point clouds with obvious shape features, the method has better recognition effect. Next, the proposed keypoint extraction algorithm is introduced. As shown in Figure 1a, this paper first adopts the method of grid downsampling, performs mean processing on the points in the grid, and combines ISS sampling on the basis of the grid curvature sampling, and finally perform angle-adaptive judgment on the extracted sampling points to generate the K-PPF key points.

#### 2.1.1. Curvature Sampling Point

Curvature is dependent on the concavity and convexity of the model. Using the Principal Component Analysis (PCA) [28] method based on normal vector estimation, the curvature of each data point can be estimated based on normal vector estimation. The curvature estimation method is as follows:

In Equation (1) [29], $\lambda_{0}$ describes the change of the surface along the normal vector, while $\lambda_{1}$ and $\lambda_{2}$ represent the distribution of data points on the tangent plane. The following formula is defined as the surface variation of the data point $H_{i}$ in the k neighborhood.
(1)$$\;\delta =\frac{\lambda_{0}}{\lambda_{0}+\lambda_{1}+\lambda_{2}}$$

The curvature $\omega_{i}$ of the point cloud model at the data point can be approximated as the surface variation δ at the point, that is $\omega_{i}\approx \delta$.

Curvature pre-search: because of the enormous number of plane features, choosing point clouds with curvatures larger than a specified value as keypoints may significantly minimize the number of duplicate point pairs. As a result, curved points can also effectively represent corners and edges of objects, improving their matching efficiency. By setting the curvature threshold $\varphi_{1}$, a large number of plane points can be removed, and an n dimensional vector P is established to store the pre-searched point $\varphi_{i}$.
(2)$$\;\varphi_{1}=\frac{1}{n}\overset{n}{\underset{i=1}{\sum }}\omega_{i}$$

Adaptive point selection: set the threshold $\varphi_{2}$ to adaptively extract the top m points with the largest curvature from P according to the point cloud number algorithm. $P_{s}$ is the number of point clouds P.
(3)$$\;m=P_{s}\varphi_{2}$$

#### 2.1.2. ISS Sampling

Intrinsic Shape Signature (ISS) [30] is a method for representing solid geometric shapes. The algorithm has a wealth of geometric feature information and is capable of completing high-quality point cloud registration. Suppose the point cloud p contains n points $\left(x_{i},y_{i},z_{i}\right),\;i=1,2,\cdot \cdot \cdot ,n-1$, set $p_{i}=\left(x_{i},y_{i},z_{i}\right)$, the specific process of extracting feature points would be follows:

A search radius $r_{seek}$ is set for each query point $p_{i}$. Then, calculate the Euclidean distance between the query point $p_{i}$ and each point $p_{j}$ in the neighborhood, and the covariance matrix $cov\left(p_{i}\right)$ between each query point $p_{i}$ and all points in the neighborhood, and set the weight $\omega_{ij}$.
(4)$$\omega_{ij}=\frac{1}{\left\|p_{i}-p_{j}\right\|}\;\|p_{i}-p_{j}\|<r_{seek}$$
(5)$$cov\left(p_{i}\right)=\frac{\sum_{\|p_{i}-p_{j}\|<r_{seek}}\omega_{ij}\left(p_{i}-p_{j}\right){\left(p_{i}-p_{j}\right)}^{T}}{\sum_{\|p_{i}-p_{j}\|<r_{seek}}\omega_{ij}}$$

Finally, calculate all the eigenvalues $\left\{\lambda_{i}^{1},\lambda_{i}^{2},\lambda_{i}^{3}\right\}$ of the covariance matrix $cov\left(p_{i}\right)$, and sort them in descending order. Set the thresholds δ1 and δ2 and satisfy the formula (6) to be the feature points.
(6)$$\;\{\begin{array}{c} \frac{\lambda_{i}^{2}}{\lambda_{i}^{1}}\leq \delta 1\\ \frac{\lambda_{i}^{3}}{\lambda_{i}^{2}}\leq \delta 2 \end{array}$$

#### 2.1.3. Angle-Adaptive Judgment

For any point $P_{i}$ in the point cloud P, let the normal of $P_{i}$ be $n_{i}$, the normal of its adjacent point $P_{ij}$ be $n_{ij}$, and define $S\left(P_{i}\right)$ as the mean of the angle between the normals of the neighborhood: where m is the number of points in the neighborhood, *n* is the number of point cloud sampling points, *j* = 1, 2, …, *m*; *i* = 1, 2, …, *n*. The value range of $S\left(P_{i}\right)$ is [0, 180], where the larger the value of $S\left(P_{i}\right)$, the larger the angle between $n_{i}$ and $n_{ij}$, as shown in Figure 2a; the smaller $S\left(P_{i}\right)$ means that the angle between $n_{i}$ and $n_{ij}$ is smaller, as shown in Figure 2b. We use the global mean of the normal angle as the threshold $\varepsilon_{1}$ as the final detection condition for keypoints. Therefore, the angle determination condition based on the mean of the normal angle is as follows: count the normal angle between each point and other points, if the 90% of the point normal deviations are above $\varepsilon_{1}$, and the sample points are reserved as the keypoint output.
(7)$$S\left(P_{i}\right)=\frac{1}{m}\overset{m}{\underset{j=1}{\sum }}\cos^{-1}\frac{n_{i}\cdot n_{ij}}{\left|n_{i}||n_{ij}\right|}$$
(8)$$\;\varepsilon_{1}=\frac{1}{n}\overset{n}{\underset{i=1}{\sum }}S\left(P_{i}\right)$$

The results of the proposed keypoint extraction algorithm are shown in Figure 3. It can be seen that most of the sampling points are distributed at the corners with large edges and curvature change information. For some scenes with sharp edges and corners, the extraction effect is more obvious. This paper’s keypoint sampling can better represent 3D objects with fewer point pairs than grid uniform sampling.

#### 2.1.4. Point Pair Feature

The point pair feature [16] uses 4 parameters to describe the relative position and direction of the two orientation points, as shown in Figure 1b; the point pair feature composed of the point pair $\left(m_{1,},m_{2}\right)$ and its normal $\left(n_{1,},n_{2}\right)$ is $F(m_{1,},m_{2})$, which is defined as a four-element vector.
(9)$$F\left(m_{1,},m_{2}\right)=\left({\left\|d\right\|}_{2},∠\left(n_{1},d\right),∠\left(n_{2},d\right),∠\left(n_{1},n_{2}\right)\right)$$
where ∠(x,y) represents the angle between the two vectors, and ${\left\|d\right\|}_{2}$ represents the distance between the point pairs.

#### 2.1.5. Global Feature Description

Based on the reference [16], we need to use point pair features to create global feature descriptors in the offline stage. For this, we need to compute the point pair features $F_{m}$ for all sample points in the model point cloud; for each sample point on the model surface $(m_{i},m_{j})\in Model$, this paper adopts keypoint extraction for sampling. Then, the similar point pair feature vectors are combined and stored under the same entry in the hash table. The whole process is the mapping from the sampling point to the feature space to the model. As shown in Figure 4, examples of point pairs with similar features on a single object are shown, which are collected in the hash table set A. During the training process, the global model description is represented as a hash table indexed by sampled features $F_{m}\left(m_{i},m_{j}\right)$, and the scene feature $F_{s}(s_{i},s_{j})$ is given as a key to access the hash table for searching in the online stage.

### 2.2. Online Stage

In the online stage, we randomly select a reference point from the keypoints extracted from the scene point cloud, and all other points in the scene are paired with the reference point to create point pair features. We obtain potential matches by matching these features with model features contained in the global model description hash table. We finally use a Hough-like voting scheme to vote on the pose of the object to return the best estimated pose for coarse matching.

#### 2.2.1. Voting Strategy

If a point pair $\left(s_{i},s_{j}\right)$ in the scene has a similar point pair feature vector to a point pair $\left(m_{i},m_{j}\right)$ in the model, it is considered that the reference point $s_{i}$ in the scene matches the point $m_{i}$ in the model, respectively. Translate the two reference points to the coordinate origin and rotate the two-point normals $\;n_{i}^{m}$, $\;n_{i}^{s}$ to the positive half-axis of the x-axis. At this time, the transformation from the model to the scene can use a rotation angle α description, the transformation from model point $m_{i}$ to scene point $s_{i}$ is defined as:(10)$$\;s_{i}=T_{s\to g}^{-1}R_{x}\left(\alpha \right)T_{m\to g}m_{i}$$

In the Equation (10) [16]: $T_{s\to g}$ represents the translation and rotation transformation from the reference point in the scene to the coordinate system; $T_{m\to g}$ represents the translation and rotation transformation from the model reference point to the coordinate system; $R_{x}\left(\alpha \right)$ represents the rotation around the *x*-axis. The positive half-axis is rotated by an angle of α.

In the online matching stage, we calculate all point pair features of the scene point cloud, find the matching model point pair and rotation angle in the hash table, and cast a vote in the corresponding position of the voting table. This scheme is similar to generalized Hough voting, as shown in Figure 1g.

#### 2.2.2. Pose Clustering

To both filter out inaccurate poses and increase the accuracy of the final findings, we cluster the recovered poses so that all poses in a cluster do not deviate in translation and rotation by more than a predetermined threshold. The score of a cluster is the total of the contained posture scores, which represent the number of votes earned in the voting system. The resultant posture is calculated by averaging the poses contained in the cluster after locating the cluster with the highest score.

## 3. Fine Registration

Because the pose produced after voting clustering may not be precise enough after K-PPF coarse registration, this study employs point-to-plane ICP for correct registration on this basis.

### Point-to-Plane ICP

Low [2] devised and extensively utilized the point-to-plane ICP method. The method takes the normal of point $q_{i}$ in the source point cloud set Q through the point $p_{i}$ in the target point cloud set P as the matching point, where the corresponding distance between the two points is the distance from $p_{i}$ to the tangent plane where $q_{i}$ is located. Using the distance from the point to the tangent plane as a measure for aligning two sets of point clouds speeds up ICP convergence and assures high alignment accuracy in complicated scenarios, but the normal vector of the point cloud must be determined beforehand. Figure 5 displays the algorithm’s basic architecture.

In the Figure 5, the bottom indicates the source point cloud, the top indicates the target point cloud, $p_{i}$ is the point on the target point cloud, $q_{i}$ is the point on the source point cloud, $l_{i}$ indicates the distance from the point on the source point cloud to the corresponding point tangent plane of the target point cloud, and $n_{i}$ indicates the corresponding normal to $p_{i}$. For the point-to-plane ICP algorithm, the error function is designed by minimizing the sum of squared distances.

## 4. Performance Evaluation Experiments

To demonstrate the advantages of K-PPF’s sampling method, we test the recognition effect of various experimental scenarios. First, the ADD-S metric [31] is used as the evaluation index in the UWA dataset to compare the recognition efficiency of the K-PPF algorithm compared to PPF in complex scenes, and then the algorithm in this paper is evaluated in the Redkitchen scene with rich corners. Recall and precision [32] are used as indicators to compare the algorithm in this paper with CSHOT [33], SHOT [8], SI [10], and FPFH [9]. Finally, the recognition effect of the algorithm in this paper is shown on the Kinect dataset. Through experiments in various different scenarios, the results prove the robustness and superiority of the algorithm in this paper, and their reliability is verified by the real engine dataset. The experimental environment is VS2017 and PCL1.8 [34]; the computer configuration is Intel i5-6300HQ 2.30 GHz CPU, NVIDIA GTX960m GPU, 16 GB memory Windows 10 system environment; all algorithms use OpenMP multi-core parallel acceleration.

### 4.1. Datesets

This paper uses thre public datasets and real engine dataset collected using realsense d435i as tests. As shown in the Figure 6, there are Cvlab’s kinect dataset [35], Princeton’s redkitchen dataset [36] and UWA T-rex dataset [37]. The Minolta VIVID 910 scanner created the UWA dataset, which comprises five model point clouds and 50 scene point clouds. The Kinect dataset includes six models and 16 scenes captured by the Microsoft kinect depth camera. Data from a complex point cloud with grid quality, mild occlusion, and clutter. The Redkitchen dataset comprises 60 point clouds, each of which is a 3D surface point cloud combined from 50 depth frames using TSDF volume fusion.

### 4.2. Comparison with Original PPF Algorithm

In this section, we will compare our method to the original PPF algorithm using various indicators in various settings.

#### 4.2.1. UWA Dataset (Complex Scene)

In order to verify the efficiency of the algorithm in this paper, on the basis of the paper [38], the average nearest point distance (ADD−S) [31] is used as the error estimator to compare the advantages of the algorithm in this paper compared with the PPF algorithm, the estimated pose $P^{\prime }=\left(R^{\prime },T^{\prime }\right),$ and the actual pose of the dataset P=(R,T), to calculate the average point ADD.
(11)$$ADD=\frac{1}{n}\underset{x\in N}{\sum }\|\left(Rx+T\right)-\left(R^{\prime }x+T^{\prime }\right)\|$$
where *n* is the number of points in the model point cloud, and on the basis of ADD, we use the average nearest point distance (ADD−S) metric to display the area under the accuracy threshold curve (AUC).
(12)$$ADD-S=\frac{1}{n}\underset{x\in N}{\sum }\|\left(Rx_{1}+T\right)-\left(R^{\prime }x_{2}+T^{\prime }\right)\|$$

In the formula, $x_{1}$ and $x_{2}$ are the two closest points in the model pose and the estimated pose. The AUC is calculated by changing the threshold of the average distance. The maximum threshold in this paper is set to 5 cm, and the data is trex in the UWA dataset. First, the algorithm in this paper is tested for different keypoint sampling thresholds d, and the AUC curves under different distance thresholds are obtained. It can be seen that as the threshold d increases, the time consumption increases, and the accuracy rate also increases. This is because the larger d, the more keypoints are extracted, so the effect is best when d=0.6, but the computational cost is too high. Finally, considering d=0.8 in this paper, the comprehensive effect is the best.

In Figure 7 and Figure 8, accuracy is defined as the ratio of successful recognitions to the total number of ambient point clouds at various distance thresholds. The recognition effects of 50 scenarios are then assessed for PPF and K-PPF. Figure 9 depicts a portion of the experimental data. The experimental results show that the algorithm in this paper can achieve better recognition effects in complex scenes with the same time consumption and higher efficiency than the original PPF, demonstrating the recognition advantages of K-PPF in general scenes. However, the recognition advantages of K-PPF in corner-rich scenes must be verified.

#### 4.2.2. Overlap Rate Calculation

One of the key indications for 3D object identification is the average closest point distance measurement. This is appropriate for situations in which the model’s true stance is known, but there may also be unknown positions in the actual setting. Therefore, this paper needs to use an overlap rate indicator. Reference [39] projects the object into the depth map and defines the overlap rate by the depth deviation of the projected pixels. If the depth deviation reaches the set threshold, the point overlaps, otherwise it does not overlap. The ratio of the number of overlapping pixels to the total projected pixels is the overlap ratio.

On the basis of reference [40], this paper proposes a new method for calculating the overlap ratio. First, the kd-tree structure of the model and the scene is established, and the scene points within the neighborhood $\varepsilon_{r}$ of a given radius of each model point are counted ($\varepsilon_{r}=5\times ppi$), where ppi is the point cloud density. If the number of points exceeds the set threshold $\mu_{r}$, the model point is considered to be coincident with the scene, and then it is judged whether the point cloud is occluded. If there is no occlusion, the overlap rate is $\varepsilon_{model}/\varphi_{model}$; if there is occlusion, output occlusion rate and occlusion rate $\left(\partial_{model}-\partial_{scene}\right)/\varphi_{model}$, and overlap rate is $\partial_{model}/\partial_{scene}$. In the formula, $\varepsilon_{model}$ is the number of overlapping points, $\varphi_{model}$ is the total number of points with the model, $\partial_{scene}$ is the number of overlapping points of the scene, and $\varphi_{model}$ is the number of overlapping points of the model.

#### 4.2.3. Redkitchen Dataset (Rich Corners)

In order to verify the recognition advantages of keypoint extraction in the K-PPF algorithm in scenes with large changes in curvature angle, we selected a building scene with rich corners for testing, and the data set used 40 sets of Redkitchen large-scale point cloud data. As shown in Figure 10, the keypoints extracted by the K-PPF algorithm differ more than the PPF algorithm, and can express the shape features of the object with fewer point pair features. We show the advantages and disadvantages of the sampling method in this paper through 40 sets of experimental data. Table 1 analyzes the average reduction of the matched point pair features in each scene. It can be seen that K-PPF has an average reduction in the scene matching process compared to PPF. Therefore, the computational cost of our algorithm is greatly reduced, and the overlap rate is also improved compared with the PPF algorithm. While buildings have sharp corners and their density varies greatly, our method can detect corners stably. Therefore, in scenes with large curvature changes, the algorithm in this paper can achieve better results in less time; in normal scenes, the algorithm in this paper can also achieve better recognition results in the same time.

### 4.3. Compare with Other Algorithms

To explain the benefits of the K-PPF algorithm more clearly, we compare it to CSHOT [31], SHOT [8], SI [10], and FPFH [9]. The PCL [34] open-source library serves as the experimental environment, and the data set consists of 40 Redkitchen groups. The standards of recall and precision we use are based on the transformed pose matrix. After matching with different algorithms, the generated pose matrix is compared with the initial pose of the model. If the xyz error is lower than the threshold φ, they are considered to match. Comparing each descriptor in the source point cloud with each descriptor in the transformed point cloud, we count the number of correct matches as well as the number of false matches. Changing the value of φ obtains the curve. The results are displayed as recall and precision, which are defined as:$$Recall=\frac{the\;amount\;of\;true\;positives}{total\;amount\;of\;ground\;truth\;loop\;closures}$$
$$Precision=\frac{the\;amount\;of\;true\;positives}{total\;amount\;of\;detected\;loop\;closures}$$

In the formula, the algorithm determines that the correct condition is that the overlap rate is greater than 80%; that is, it is considered a successful match. It can be seen from Figure 11 below that under different thresholds, the recall rate and precision rate of the algorithm in this paper are compared with the other four algorithms. For better results, the recall rate is 12.5% higher than CSHOT and 100% higher than SI.

### 4.4. Point Cloud Recognition Experiment

To verify the point cloud recognition effect of the algorithm in this paper in complex environments and different degrees of occlusion, the kinect data set of CVlab is used to test 5 sets of data, including 5 model point cloud data and 18 scene point cloud data. As shown in Figure 12, the red bounding box is drawn for the model point cloud identified in the scene point cloud, and the point cloud recognition effect is displayed in green. It can be seen from the recognition results in Table 2 that in the 5 groups of different degrees of occlusion, the average data overlap rate is above 90%.

### 4.5. Real Dataset Experiment

In order to verify the reliability of the algorithm, after preprocessing the real scene point cloud collected by the depth camera and the engine CAD model point cloud in the laboratory, a comparison between the algorithm in this paper and the original PPF algorithm is made. As shown in Figure 13, it can be seen that the initial poses of the two point clouds are quite different, and the shapes are not exactly the same. In this case, the registration effect of the algorithm is good. Table 3 displays the findings. When compared to the PPF algorithm, our method reduces redundant point pairs by 17.4%, takes 33.3% less time, and increases the overlap rate by 1.731%.

## 5. Conclusions

Aiming at the shortcomings of original PPF grid sampling redundant point pairs, this paper proposes a keypoint extraction algorithm based on grid ISS sampling combined with curvature-adaptive sampling and angle-adaptive judgment (K-PPF), which have the characteristics of high efficiency and strong robustness. The algorithm improves the sampling part of the original PPF and contains less duplicate point pairs. Finally, the efficiency of the method in this study is validated by comparison with other algorithms, and the program’s recognition accuracy in the complex environment and occlusion process is validated using the CVLab dataset. In practical applications, the proposed keypoint algorithm is easy to match and can provide a good initial pose for matching. The experimental results of real datasets also prove that the algorithm in this paper has very high efficiency and can be used for laboratory virtual–real fusion experiments in the future. 

Howerver, although the paper improves the real-time performance of 3D point cloud feature extraction, it still needs further improvement for large-scale and real-time application scenarios. In the future, innovations in voting strategies can be made to reduce the time consumed by algorithms.

## Figures and Tables

**Figure 1 sensors-22-06289-f001:**
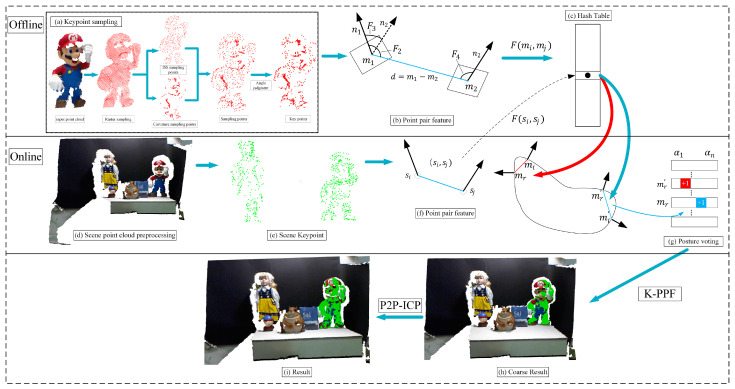
The algorithm flow of this paper; first, in the offline stage, the model training keypoint pair features are stored in the hash table (**a**–**c**). Then, the online stage preprocesses the scene point cloud (**d**), extracting keypoint pair features and performing the model point cloud hash table Quickly vote (**e**,**f**), clustering the point clouds with high votes (**g**,**h**); finally, point-to-plane ICP for further fine registration is performed, and the 6D pose after fine registration derived (**i**).

**Figure 2 sensors-22-06289-f002:**
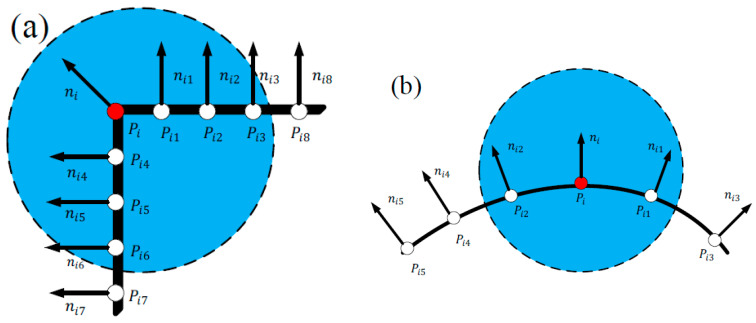
Shown from different angles. (**a**) corner point; (**b**) surface point.

**Figure 3 sensors-22-06289-f003:**
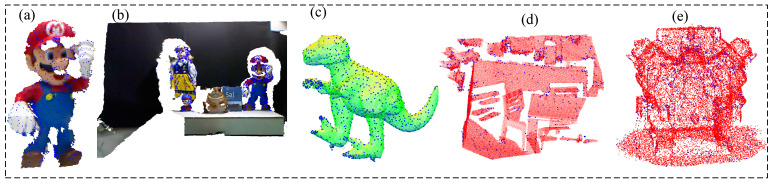
Demonstrate keypoints of K-PPF (**a**) kinect Mario model; (**b**) Kinect scene; (**c**) UWA Trex; (**d**) Redkitchen; (**e**) our lab’s engine.

**Figure 4 sensors-22-06289-f004:**
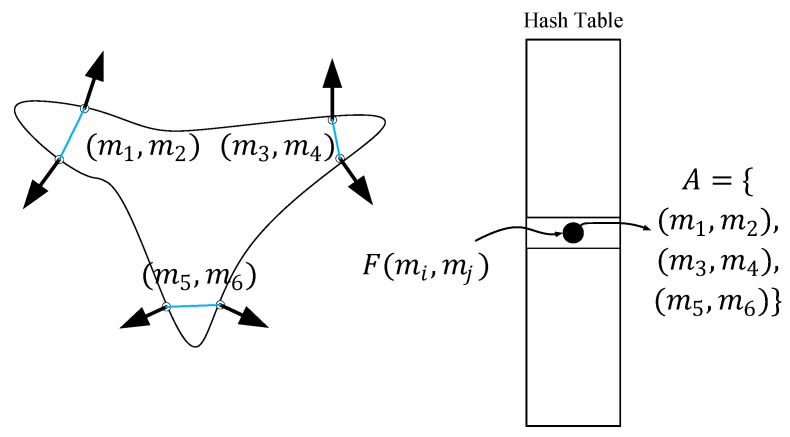
Model point pairs with similar features.

**Figure 5 sensors-22-06289-f005:**
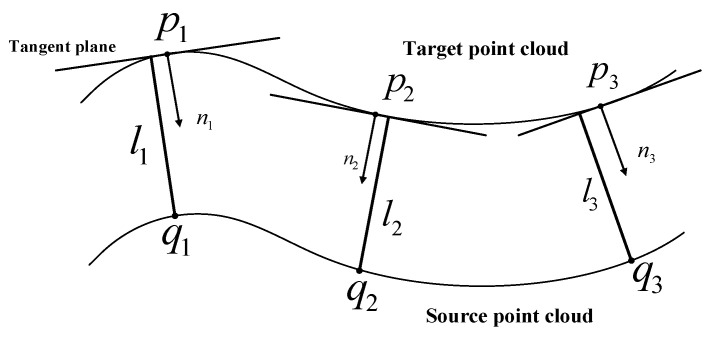
Diagram of the point-to-plane ICP algorithm.

**Figure 6 sensors-22-06289-f006:**
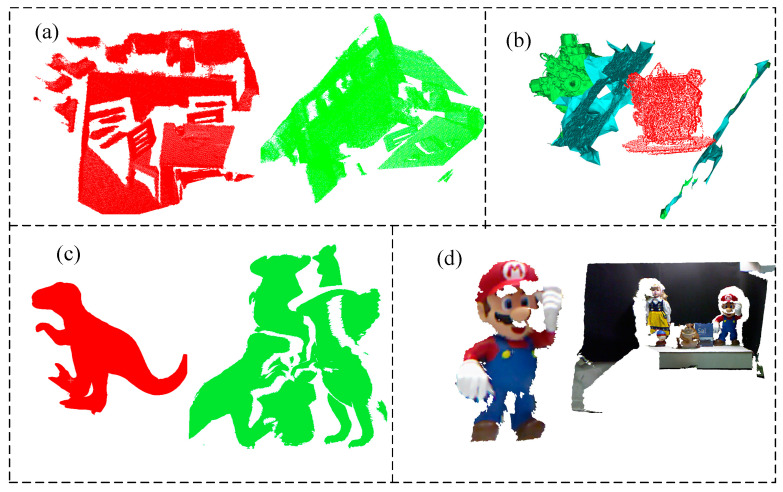
(**a**) Redkitchen dataset; (**b**) Engine; (**c**) UWA dataset: T-rex model and scene; (**d**) Kinect dataset: Mario and scene.

**Figure 7 sensors-22-06289-f007:**
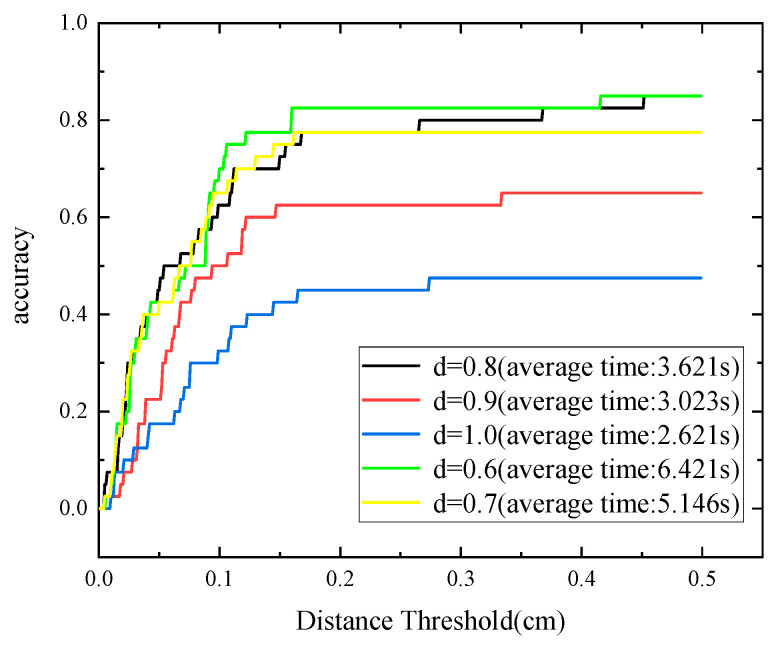
Distance threshold–accuracy curve.

**Figure 8 sensors-22-06289-f008:**
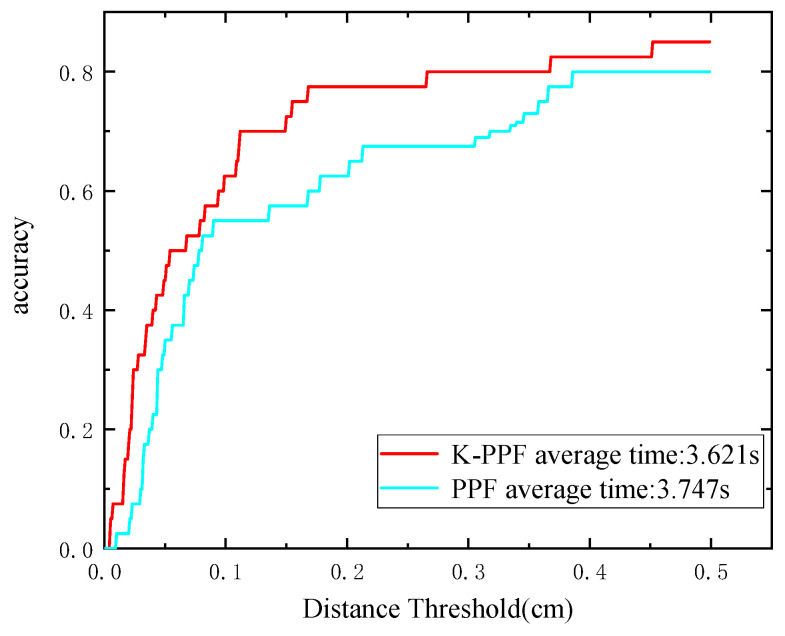
AUC comparison of K-PPF and PPF.

**Figure 9 sensors-22-06289-f009:**
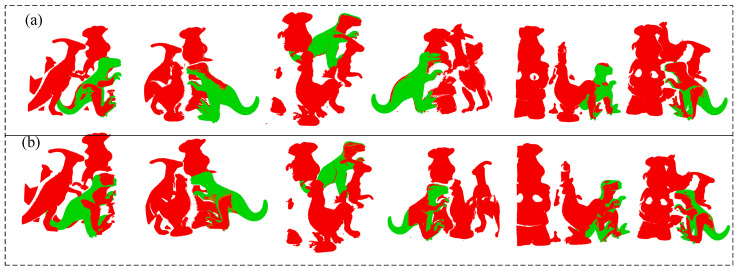
Matching effect display. (**a**) K-PPF; (**b**) PPF.

**Figure 10 sensors-22-06289-f010:**
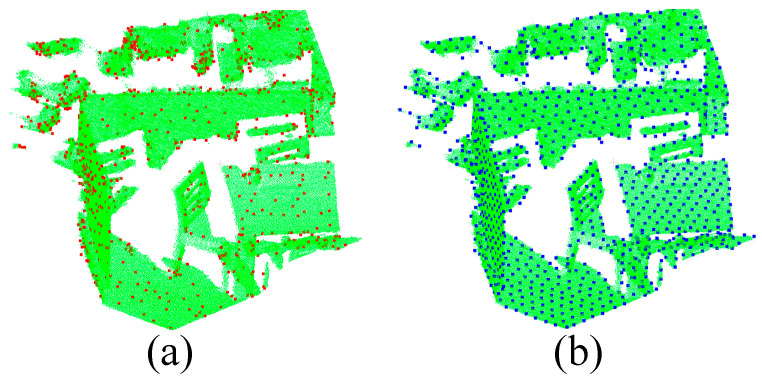
Sampling point comparison display (**a**) K-PPF, red is the sampling point; (**b**) PPF, blue is the sampling point.

**Figure 11 sensors-22-06289-f011:**
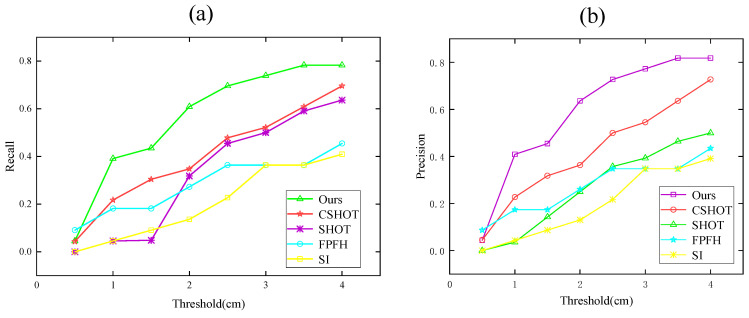
Algorithm comparison; (**a**) Recall; (**b**) Precision.

**Figure 12 sensors-22-06289-f012:**
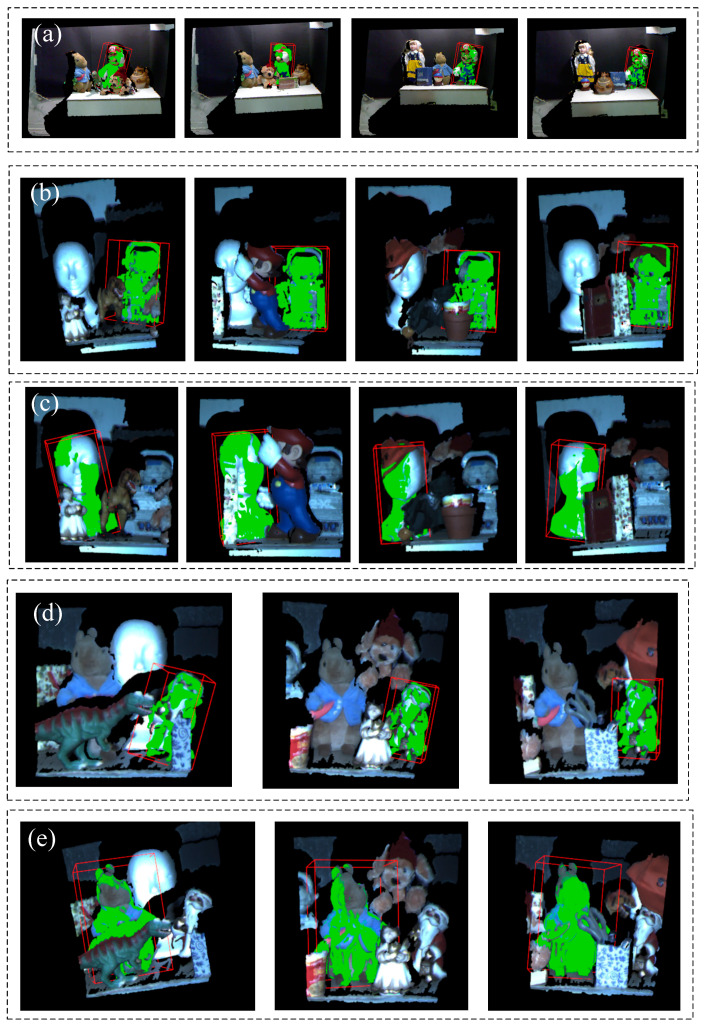
5 sets of matching situations in different scenario. (**a**) Mario; (**b**) robot; (**c**) face; (**d**) doll; (**e**) Peter Rabbit.

**Figure 13 sensors-22-06289-f013:**
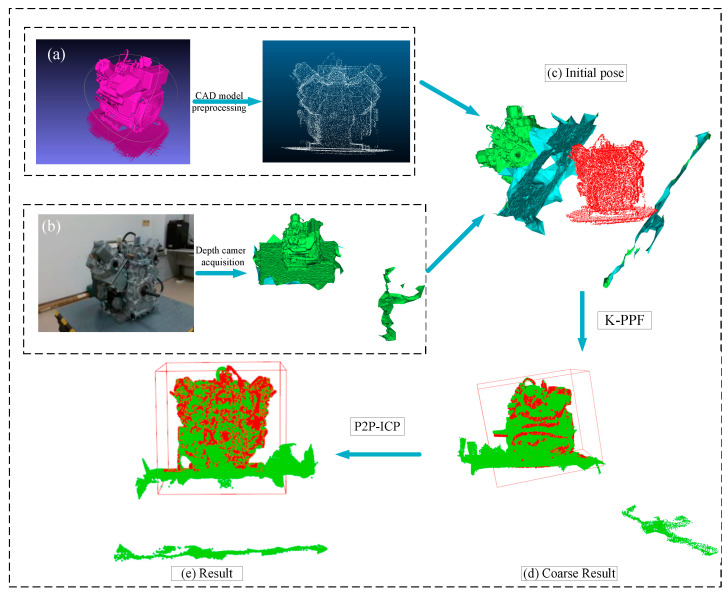
Real dataset recognition effect. (**a**) CAD model downsampling; (**b**) Scene point cloud collection; (**c**) Initial pose; (**d**) K-PPF coarse registration result; (**e**) Fine registration result.

**Table 1 sensors-22-06289-t001:** Comparison of algorithms: PN is the average number of scenes matching point pair features; PR is the average number of matching point pair features reduced in each scene by K-PPF compared to PPF.

Method	PN	PR	Overlap Rate/%	Time/s
PPF	98	0	95.314	5.24
K-PPF	71	27	96.1419	3.71

**Table 2 sensors-22-06289-t002:** 5 sets of data recognition results.

Data	Average Overlap Rate/%	Average Occlusion Rate/%
(a)	93.2966	22.01925
(b)	92.2091	27.9866
(c)	91.5807	28.6893
(d)	90.7508	32.4563
(e)	96.214	15.671

**Table 3 sensors-22-06289-t003:** Compared with original PPF. PN is the average number of scenes matching point pair features.

Method	PN	Overlap Rate/%	Time/s
PPF	190	97.02	1.791
K-PPF	167	98.733	1.194

## Data Availability

Not applicable.

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
