# Peer review of "A Fast Point Cloud Recognition Algorithm Based on Keypoint Pair Feature"

_sensors, 2022, doi:10.3390/s22166289_

Round 1

Reviewer 1 Report

The paper introduces a K-PPF method for 6-DOF pose estimation from point cloud scenes. The main improvement is a new sampling strategy based on additional geometric cues such as keypoints and curvatures. The method compared favaroublly to the original PPF method and some descriptor based methods.

Merits:

1. Clear writing and clear figure illustrations.

2. The experiments are were designed.

Demerits

1. The novelty is a little bit limited as many works have been down for the PPF smart sampling stratgies. The authors should clarify the distinctiveness.

2. It will be better to compare more PPF variants in the experiments.

Author Response

Thank you for your comments, I will seriously revise.

Reviewer 2 Report

The work deals with the problem of recognizing and register 3D items in various postures, as well as acquire precise and reliable 6D locations of objects using 3D data. The paper proposes a 6D pose estimation method based on Keypoint Pair Feature (K-PPF) voting, and an improved algorithm for the sampling point part. Throughout the article the original contribution is not clearly expressed. As an example:

1.      in the "proposed method" section, lines 109, 110 the division of the algorithm into offline and online stages is proposed, similarly to other papers, without any reference: 

a.      1 Drost, B.; Ulrich, M.; Navab, N.; Ilic, S. Model globally, match locally: Efficient and robust 3D object recognition. In Proceedings 464 of the IEEE Computer Society Conference on Computer Vision and Pattern Recognition, San Francisco, CA, USA, 13–18 June 465 2010, 

b.      M. Pauly, M. Gross and L. P. Kobbelt, "Efficient simplification of point-sampled surfaces," IEEE Visualization, 2002. VIS 2002., 2002, pp. 163-170, doi: 10.1109/VISUAL.2002.1183771

2.      In paragraph 3.1.1, line 139, surface variation is introduced without reference (formula (1)) [M. Pauly, M. Gross and L. P. Kobbelt, "Efficient simplification of point-sampled surfaces," IEEE Visualization, 2002. VIS 2002., 2002, pp. 163-170, doi: 10.1109/VISUAL.2002.1183771]

3.      In Point Pair feature formula (9) is introduced without any reference

4.      In the experimental part, the authors compare the proposed algorithm with CSHOT, SHOT, SI and FPFH. Which implementation of the algorithms did they use? Did they make use of specific libraries such as PCL?

To assess the validity of the algorithm, comparisons were made of their method to the original PPF algorithm using various indicators in various settings, with public domain datasets. However, no comparison was performed on the real dataset. At least a comparison with the original PPF algorithm should be performed.

The paper may be accepted with improvements.

1)     Throughout the article recheck punctuation

2)     Thorough English check is required

3)     the introduction does not highlight the authors' work and progress in the research, which are instead described in the 'Related work' section. “Introduction” and “Related work” should be collapsed in a single chapter.

4)     The improvement achieved by the proposed algorithm is partially stated in the abstract but not in the conclusions

Author Response

Thank you very much for your suggestion, I have seriously revised the article.

Round 2

Reviewer 1 Report

No further concerns